# A New Experimental Design to Examine Cognitive Biases for Gastrointestinal Related Stimuli in Children and Adolescents

**DOI:** 10.3390/children10081327

**Published:** 2023-08-01

**Authors:** Ellen Bjerre-Nielsen, Karen Hansen Kallesøe, Eva Skovslund Nielsen, Tine Bennedsen Gehrt, Lisbeth Frostholm, Charlotte Ulrikka Rask

**Affiliations:** 1Research Unit, Child and Adolescent Psychiatry, Aarhus University Hospital Psychiatry, Palle Juul-Jensens Boulevard 175, 8200 Aarhus, Denmark; 2Department of Clinical Medicine, Faculty of Health, Aarhus University, Palle Juul-Jensens Boulevard 99, 8200 Aarhus, Denmark; 3Center on Autobiographical Memory Research, Department of Psychology and Behavioural Sciences, Aarhus BSS, Bartholins Allé 11, 8200 Aarhus, Denmark; 4Department of Research and Development, Prehospital Emergency Medical Services, Central Denmark Region, Brendstrupgårdsvej 7, 8200 Aarhus, Denmark; 5The Research Clinic for Functional Disorders and Psychosomatics, Aarhus University Hospital, Palle Juul-Jensens Boulevard 11, 8200 Aarhus, Denmark

**Keywords:** adolescent, children, cognitive bias, functional abdominal pain disorder, functional gastrointestinal disorders, interpretation bias, memory bias

## Abstract

Cognitive biases toward disorder-specific stimuli are suggested as crucial to the development and maintenance of symptoms in adults with functional gastrointestinal disorders (FGID). Functional abdominal pain disorders (FAPD), a subtype of FGID, are common in children and adolescents, but the influence of cognitive biases is sparsely examined. This study aimed to (1) develop a new experimental design for assessing cognitive biases toward gastrointestinal stimuli in children and adolescents (aged 8 to 17 years) and (2) derive comparative data on bias toward gastrointestinal stimuli using a healthy “normative” sample. The online experimental design–BY-GIS (Bias in Youth toward GastroIntestinal-related Stimuli)—includes a word task and a picture task. Stimuli in both tasks are related to general and gastrointestinal symptoms, and the design includes three phases: (1) encoding, (2) free recall, and (3) recognition. Data were collected between April 2022 and April 2023 from 96 healthy participants (M_age_ = 12.32, 47.92% female). Adolescents were significantly better at recalling words than children (*p* = 0.03), whereas there were no significant gender or age differences with regard to recalling pictures (*p* > 0.05). Across age and gender, participants performed above chance level in the recognition phases of both tasks. The results support that the design is suitable within the age span.

## 1. Introduction

Cognitive biases refer to abnormalities in attention to, interpretation of, and memory of specific stimuli, for instance, showing selective attention to negative information [1]. These biases are suggested to influence the development, maintenance, and recurrence of symptoms in various disorders such as anxiety, depression, and functional disorders (FD) [1,2,3]. The latter is an umbrella term for disorders characterized by certain patterns of somatic symptoms that cannot be explained by a well-defined organic or psychiatric disorder [4,5]. The etiological mechanisms of FDs are suggested to be multi-factorial, involving both the body and brain [4,5], including a symptom-focused attentional bias [2] and a lower threshold for detecting symptoms [6].

Cognitive biases can be explored in experimental studies, where participants’ attention to, interpretation of, or memory of specific stimuli (e.g., pictures or words) are assessed. A range of tasks using these principles exist, for example, the dot probe task [7], the Health Norms Sorting Task, [8] or the Implicit Association Task [9]. In patient populations, biases are often present for disorder-specific stimuli [10]. For instance, patients with irritable bowel syndrome (IBS), a subtype of FD, show attentional biases specifically toward gastrointestinal-related words, whereas patients with anxiety show attentional bias toward threat-related words [3,11]. Not only is the theme of the stimuli important, but the best type of stimuli may also depend on the disorder being explored. In chronic fatigue syndrome, another subtype of FD, studies have found attentional bias toward word stimuli but not for pictorial stimuli [10]. In contrast, pictorial stimuli have been suggested to be more favorable for assessing cognitive biases in studies on health anxiety [12,13].

A common limitation in most prior studies is that biases in attention, interpretation or memory are examined separately, although it has been suggested that they interact. Hirsch et al. proposed the combined bias hypothesis, suggesting that cognitive biases at different levels interact to maintain symptoms of a disorder [14]. Thus, examining cognitive biases together rather than in isolation may mimic how these processes unfold in everyday life and inform how the combined effect of these biases influences the development or maintenance of disorders [1,14].

So far, most experimental studies on cognitive biases in FDs have been performed on adults and have, for example, demonstrated cognitive biases toward health-related or negative stimuli in adults with IBS and functional neurological disorder [11,15,16,17]. In contrast, the research on cognitive biases in children and adolescents is scarce, although FDs are also prominent in these age groups. One of the most prevailing FDs in youth is functional abdominal pain disorder (FAPD) [18], in which IBS is included, with an estimated prevalence of more than 10% [19]. At present, the most evidence-based treatment for FAPD is cognitive behavioral therapy, but the effects are mostly slight to moderate [20,21]. Thus, increased understanding of cognitive biases in these disorders may be crucial to further optimizing psychological treatment. To date, only a few studies focusing on abdominal pain and cognitive biases in children and adolescents have been performed. Results from these studies point toward an attentional bias toward pain-related stimuli being associated with abdominal pain [22,23,24,25]. Further, Lau et al. found indication of the presence of attentional, interpretational, and memory biases in children and adolescents with chronic pain (including abdominal pain) in their review [26]. However, the authors emphasized the need for more research in which future studies include age-appropriate measures that could target different biases in samples with wide age spans in order to explore the direction and role of these biases [26]. 

The present study introduces a novel experimental design overcoming the limitations of prior studies by including both word and picture stimuli to examine interpretation and memory biases regarding gastrointestinal-related stimuli in children and adolescents. Further, this new design includes the potential to assess the interplay of biases in interpretation and memory. Specifically, the aims are: (1) to develop a new experimental design, “Bias in Youth toward GastroIntestinal-related Stimuli” (BY-GIS), for assessing cognitive biases toward gastrointestinal-related stimuli in children and adolescents, and (2) to explore potential gender- and age-related differences and derive initial comparative data on bias regarding gastrointestinal-related stimuli using a healthy “normative” sample. 

## 2. Materials and Methods

### 2.1. Study Design and Procedure

This is a cross-sectional study with online data collection carried out from April 2022 to April 2023 using REDCap (Research Electronic Data Capture) [27,28]. Potential participants, i.e., healthy children and adolescents, were recruited via social media, school intranets, and word of mouth. Information regarding eligibility according to predefined study in- and exclusion criteria was provided by a parent and/or the participant (if aged ≥ 15 years) in a telephone interview. If the participant met the in- and exclusion criteria, written informed consent was obtained from parents in participants <15 years and from a parent and the participant in participants aged ≥15 years. The informed consent form was sent via a secure and personal email system (DigitalPost) and stored in REDCap [29]. All children and adolescents who completed the study took part in a great dealtery to win a tablet or Bluetooth headphones.

Inclusion criteria were age between 8 and 17 years and Danish language proficiency. Exclusion criteria were (1) chronic physical diseases or disabilities demanding treatment or follow-up in a hospital-setting, (2) a psychiatric diagnosis, (3) current psychological treatment or (4) experiencing abdominal pain more than twice during the past two weeks [23].

### 2.2. Experimental Design

BY-GIS included a word task and a picture task, both developed to assess the possible interplay of cognitive biases in interpretation and memory for each type of stimuli. BY-GIS consisted of three phases: (1) encoding, (2) free recall, and (3) recognition. The order of presentation of the phases was word encoding, picture encoding, word free recall, picture free recall, word recognition, and picture recognition. The order ensured a short time interval between each phase concerning the same stimuli. Hence, the picture tasks served as filler tasks for the word tasks and vice versa. Both word stimuli and picture stimuli were included because they had different targets. The words targeted specific gastrointestinal symptoms, while the pictures showed different situations that are typically affected when children and adolescents experience abdominal pain (e.g., social situations with peers involving food) [30]. For an overview of the BY-GIS design, see Figure 1. BY-GIS was pilot-tested on six children and four adolescents, which resulted in minor changes to the set-up to shorten test time (i.e., removal of a task to describe each picture in the encoding phase).

#### 2.2.1. Word Task

The word task was inspired by the Health Norms Sorting Task [8], which has previously been used to assess cognitive biases in children and adults [32,33]. In the encoding phase, participants were presented with a list of physical symptoms and asked to attend to one symptom at a time while imagining developing that symptom. For each word, participants indicated whether they would perceive themselves as healthy or no longer healthy if they developed the symptom. Two categories of symptoms were included in the present study: (1)general symptoms (e.g., coughing, headache);(2)gastrointestinal symptoms (e.g., nausea, constipation)

Each category comprised 10 words. General symptoms were selected from the Children’s Somatic Symptom Inventory and the Health Norms Sorting Task, while the gastrointestinal symptoms were selected from the Pediatric Quality of Life Inventory Gastrointestinal Symptom Scale (for a list of all included words, see Appendix A) [34,35]. In the free recall phase, participants wrote down as many symptoms as they remembered from the encoding phase. In the recognition phase, participants were presented with all symptoms from the encoding phase alongside 20 new symptoms not previously presented. The new symptoms were 10 general symptoms (from the Children’s Somatic Symptom Inventory and Health Norms Sorting Task) and 10 gastrointestinal symptoms (determined by the author group). The symptoms, both previously presented and new, were split into two lists corresponding to their category (general or gastrointestinal) and presented on separate pages. For each symptom (original and new), participants indicated whether they had seen it before or not. Further, they rated the difficulty of making this choice and the confidence in their answer on a scale from 1 to 7 (1 = not at all, 7 = a great deal) for each of the two lists of symptoms.

#### 2.2.2. Picture Task

The design of the picture task followed previous studies [36,37] but was adjusted to be suitable for a younger age group by simplifying the task. The stimuli were 15 pictures from the Pictures with Social Context and Emotional Scenes (PiSCES) database [31]. Pictures from the database have previously been used in a study on children [38]. The pictures represented three categories of social contexts (school, fun and play, and food) with five pictures in each category (see Appendix A for a description of each picture and reference number in the PiSCES database). In the encoding phase, participants saw one picture at a time and had to rate the picture on three dimensions before they could move on to the next picture, in line with previous studies [36,37]. Participants rated each picture on:(1)whether it was positive or negative (emotional valence, 0 = negative, 100 = positive);(2)if they experienced any bodily symptoms or sensations when seeing the picture (physical arousal, 1 = not at all, 7 = a great deal);(3)if the picture reminded them of something from their own life (self-relevance, 1 = not at all, 7 = a great deal).

In the free recall phase, participants were instructed to describe as many pictures as they could remember from the encoding phase. Each remembered picture was described and rated separately on emotional valence and physical arousal using the same questions as in the encoding phase. In the recognition phase, participants were presented with all the pictures from the encoding phase along with the picture’s mirror image. Each original picture and mirror image pair was presented separately. Participants indicated which of the two pictures they had seen in the encoding phase, rated how difficult it was to choose between them (difficulty, 1 = not at all, 7 = a great deal), and how confident they were in their choice (confidence, 1 = not at all, 7 = a great deal).

### 2.3. Additional Measures

Parents provided information on sociodemographic background, i.e., their education, employment, and cohabitant status. The demographics of the participants (age and gender) were obtained when participants signed up for the study. After completing the experimental tasks, participants answered three questionnaires on nonspecific somatic complaints, illness-related anxiety, and quality of life (see below).

(1)The Children’s Somatic Symptoms Inventory, formerly known as the Children Somatization Inventory, consists of 24 items assessing nonspecific somatic symptom complaints in children [35]. The questionnaire is rated on a 5-point Likert scale from 0 to 4 (0 = not at all, 4 = A whole lot).(2)The Childhood Illness Attitudes Scale assesses symptoms of health anxiety in school children [39] and is a modified version of the Illness Attitudes Scales. The Childhood Illness Attitudes Scale has 35 items and is rated on a 3-point Likert scale ranging from 1–3 (1 = none of the time, 3 = a lot of the time), with a higher score indicating a higher level of health anxiety. In the present study, only the “fears” subscale (11 items) was included [40].(3)The Pediatric Quality of Life Inventory, Gastrointestinal Symptom Scale measures symptoms related to functional gastrointestinal disorders [34]. The scale has nine items rated on a 5-point Likert scale from 0–4 (0 = never a problem, 4 = almost always a problem). Items are reverse-scored; thus, a higher score indicates a better quality of life [34].

### 2.4. Power Analysis

The parameters for the power analysis were based on a previous study by Jungmann and Witthöft [32]. We assumed a power (β) of 0.8 and α = 0.05, comparing 60 patients in a clinical group with 100 healthy controls. Based on these parameters, a power analysis using the effect of two independent means gives a minimal detectable effect size of Cohens d = 0.46. Since a Cohens d = 0.4 is a typical effect size in psychological research, the estimated effect size of 0.46 was deemed adequate [41]. Thus, we aimed to recruit approximately 100 healthy participants in the current study to obtain comparison data for future studies on children and adolescents with gastrointestinal symptoms.

### 2.5. Data Analysis

Participants’ descriptions of words and pictures in the free recall phase were coded according to their correspondence to the individual words and pictures presented in the encoding phase, as well as their correspondence to a category (words: general symptoms, gastrointestinal symptoms; pictures: school, fun and play, food). If a description did not match an individual picture or word or did not match a category, it was coded as “other”. Responses from the first 20 individuals (only 19 for pictures) who completed the experimental tasks were coded by two independent raters (1st author and 3rd author), with high interrater agreement for both words (individual words: 95%, word category: 98%) and pictures (individual pictures: 98%, picture category: 100%). Disagreements between the two coders were resolved via discussion under supervision of an experienced rater (4th author). The remaining responses were coded by the first author. 

#### Statistical Analysis

The statistical analyses were performed using STATA version 17.0 MP—Parallel Edition [42]. We used Student’s *t*-test and the Wilcoxon rank-sum test to explore potential gender- and age-related differences in the outcomes of the encoding, recall, and recognition phases. All *p*-values were two-tailed and considered significant if <0.05.

## 3. Results

In total, 203 records were assessed for eligibility, and the final sample consisted of 51 children (8 to 12 years) and 45 adolescents (13 to 17 years). For more details, see Figure 2. Characteristics of participants and their parents are reported in Table 1.

### 3.1. Word Task

Table 2 shows the descriptive statistics for the three phases of the word task. 

#### 3.1.1. Encoding Phase

Participants replied that they would still be “Healthy” if they developed the majority of the symptoms in the task (median (Mdn) = 12.50, interquartile range (IQR) = 5.00). The symptom most often evaluated as “No longer healthy” was “Vomiting”. There was no significant difference between gender or age groups. 

#### 3.1.2. Free Recall Phase

The median for recalled words was 7.00 (IQR = 6.00, possible range 0 to 20). The most often recalled word was “Stomachache”, which was recalled by 81.3% of the participants, followed by “Headache”, which was recalled by 72.9% of the participants. Adolescents were significantly better than children at recalling general words (Mdn = 3.00 vs. 4.00, *p* < 0.01) and words in total (Mdn 6.00 vs. 8.00, *p* = 0.03), but not gastrointestinal words.

#### 3.1.3. Recognition Phase

The median for correctly recognized words was 37.00 (IQR = (2.00, possible range 0 to 40). This was above chance level, which corresponded to only 20 words. The adolescents were significantly better at correctly recognizing gastrointestinal words compared to children (Mdn 18.00 vs. 19.00, *p* = 0.03).

### 3.2. Picture Task

Table 3 shows the descriptive statistics for the three phases of the picture task. 

#### 3.2.1. Encoding Phase

The total sample evaluated the pictures as slightly positive (Mdn = 69.00, IQR = 11.40, range 0 to 100) with no significant difference between gender or age groups in the rating of emotional valence. There was a significant difference between children’s and adolescents’ evaluations of physical reaction, as children had a higher median compared to adolescents (Mdn = 1.47 vs. 1.20, *p* = 0.04). 

#### 3.2.2. Free Recall Phase

The median for recalled pictures was 3.00 (IQR = 2.00, possible range 0 to 15) with no significant difference between gender or age groups. The most frequently recalled picture depicted a birthday, which was recalled by 54.17% of the participants, followed by a picture depicting people in a library, which was recalled by 51.04% of the participants. There was a significant difference between girls and boys in the rating of emotional valence, as boys evaluated the recalled pictures as more positive than girls (Mdn = 64.50 vs. 74.38, *p* = 0.01). Similar to the encoding phase, children had a significantly higher median for physical reaction compared to adolescents (Mdn = 1.60 vs. 1.10, *p* = 0.04).

#### 3.2.3. Recognition Phase

Almost all 15 pictures were correctly recognized by the participants (Mdn = 15.00, IQR = 0). There was no significant difference between gender or age groups.

## 4. Discussion

To our knowledge, BY-GIS is the first experimental design specifically developed for children and adolescents to assess cognitive biases toward gastrointestinal stimuli in interpretation and memory and their interplay. This design can potentially overcome some of the limitations in prior research on cognitive biases in young people, such as small samples, lack of control group, or a narrow age span [22,25,26]. Only one out of 97 participants did not complete BY-GIS once started. Additionally, 98% of participants recalled at least one word or picture and for both recognition phases the participants performed above chance level. Further, we found few significant differences between gender and age groups across the three phases in both the word task and the picture task. Together, a healthy normative sample with comparative data on bias to gastrointestinal stimuli has been derived in children and adolescents aged 8 to 17 years.

Given the scarcity of existing literature on cognitive biases in children and adolescents with functional disorders, the development of BY-GIS was one step toward gaining more knowledge on this topic. Two previous studies assessed attentional biases in children with recurrent abdominal pain and found evidence of such a bias [22,25]. However, both studies had limitations, as Boyer et al. did not include a control group, and Hermann et al. had a small sample size (30 participants). Thus, there was a need for studies with larger sample sizes and inclusion of a healthy control group to further support the evidence of cognitive biases in children and adolescents with functional disorders. This first testing of BY-GIS overcame some of these challenges by obtaining initial data on a healthy sample that can be used for comparison in later studies on clinical samples. The healthiness of the sample was reflected in the median scores of the questionnaires as well as in the results of the encoding phase of the word task. Thus, the participants on average evaluated 37% of the symptoms as “No longer healthy”, while the sample of children and adolescents in Jungmann and Witthöft on average rated 56% of symptoms as “No longer healthy” in their version of the Health Norms Sorting Task. Participants in this study had at least one medically unexplained symptom within the last six months. Having these medically unexplained symptoms was associated with having an illness-related self-concept, i.e., a cognitive bias [32], which could in part explain the higher number of symptoms evaluated as “No longer healthy” compared to the number in our healthy sample.

In the current study, adolescents were significantly better at recalling general words and correctly recognizing gastrointestinal words compared to children. Studies on cognitive biases in adults with IBS found attentional bias when the performance of the participants distinguished specifically for one category of stimuli, i.e., gastrointestinal-related words compared to other categories, for instance, faster response time for IBS-related words than neutral words in a Stroop task [17,43]. As the better performance of adolescents in the present study was not specific for one category, we believe that the results are not due to potential bias. Instead, such a difference could be a developmental aspect which is supported by a study on free recall in eyewitness performance finding adolescents recalling more details about a film clip than children [44]. 

In the encoding phase, the overall emotional valence of the pictures was evaluated as slightly positive (Mdn = 69.00, IQR = 11.40, range: 0 to 100), in line with the original ratings in the PiSCES database [31]. The rating of physical reaction was significantly different in children and adolescents in both the encoding and the recall phases, since the children tended to rate their physical reaction higher than adolescents. This could be explained by children perceiving the pictures as more emotional than the adolescents, since research has shown that children find emotional pictures more arousing compared to neutral pictures [45]. However, for both age groups the ratings of physical reaction were low (<2, range 1 to 7), and the difference on emotional valence was not significant between age groups. 

Regarding gender, we found significant differences between girls’ and boys’ rating of emotional valence on pictures in the recall phase. The median emotional valence of the recalled pictures was lower for the girls compared to the boys. This is not an unusual finding. In a study by Belmon et al., children aged 4 to 6 years rated pictorial stimuli on a 3-point scale (negative, neutral, positive). They found that boys tended to rate the pictorial stimuli as more positive than girls [46]. Similar results were found in a Spanish study on word stimuli in children aged 7 to 13 years, where boys had higher ratings of valence than girls [47]. Additionally, the difference could be explained by the girls recalling more pictures within the “Food” and “School” categories, which included both positive, neutral, and negative pictures (as attributed in the PiSCES database [31]). In comparison, the boys recalled more pictures within the “Fun and play” category, which only contained positive pictures. 

BY-GIS included both word and pictorial stimuli, as they each displayed different targets. The word stimuli targeted specific symptoms related to FAPD, such as nausea and stomachache, since prior studies on adults with IBS have found evidence for cognitive biases specifically for IBS-related words [11,17]. In contrast, the pictorial stimuli targeted the effects of the symptoms on daily life, for instance, the fear of having a stomachache at a birthday party if you eat cake. Carlson et al. have specifically found that symptoms perceived to be induced by food negatively affected quality of life in children with functional gastrointestinal disorder [30]. Additionally, a prior study from Heathcote et al. found that youth who catastrophized about pain and had recent experiences of pain exhibited more negative interpretations of ambiguous situations, not only related to pain and bodily threat, but also toward social situations [48]. This underpins the relevance of the situations depicted in the picture task.

The recall of words and pictures as well as the ceiling effects in the recognition phases confirmed that participants paid attention throughout the task. The participants performed better in some phases, depending on the stimuli. For the recall phase, more words than pictures were recalled (35% vs. 22%). However, in the recognition phase, more pictures than words were correctly recognized (90% vs. 99%). Based on our results, it was not possible to conclude whether word stimuli were preferable over picture stimuli or vice versa. Preferably, stimuli should depend on the clinical sample [10], thus future studies on clinical samples of young people with FAPD or other disorders with gastrointestinal symptoms are needed to explore this further. Such a clinical sample would also be essential in assessing the possible interplay between cognitive biases. Three phases were included in BY-GIS in order to make it possible to assess the potential interplay of biases as previously demonstrated by Hirsch et al. in social phobia [14]. In future studies, potential associations between recall and recognition of the different stimuli categories should be evaluated. For instance, do children with FAPD more often falsely recognize new gastrointestinal words as having been previously presented? Thus, a prior study has shown that adults with IBS have a high false positive scores for recognizing negative words [49]. Additionally, Witthöft et al. also found that individuals with pathological health anxiety had a response bias toward illness and symptom words in a recognition task compared to depressed and healthy control groups [50]. 

### Strengths and Limitations

Compared to other studies on cognitive biases in children and adolescents, our study had a larger sample size and covered a wider age span [22,25,32]. Although the sample was not evenly distributed across all age groups, representation of all ages between 8 and 17 years in both genders was ensured. Further, BY-GIS included three phases (encoding, free recall, and recognition), making it possible to explore the potential interplay of biases in interpretation and memory. 

The study also had limitations. First, BY-GIS is designed to target cognitive biases in children and adolescents with abdominal pain, but in this first test, only a healthy sample was included. The lack of a clinical sample was an obvious limitation, as the feasibility of BY-GIS needs to be assessed in the proposed target group. Second, our study population was healthier and more socioeconomically advantaged than the general population. All participants were screened for current physical and mental disorders to limit the possibility of cognitive biases in relation to the employed stimuli, given the aim to derive comparison data for future studies on clinical samples. It was essential to screen our participants as a prior study found evidence of a negative interpretation bias in a general school population [48]. Therefore, there was a risk of detecting biases if we included a general school population instead of healthy sample. The parents of the participants differed from the general population in terms of education, employment, civil status, and yearly household income. For the majority of the participants, their parents lived together, had a higher education, were employed and had a yearly household income above average. Thus, our results may reflect performance on the BY-GIS of children of resourceful parents, which tend to perform better than the average child. Further, the study was conducted as an online study to make participation easier. This might be a limitation, since participants completed BY-GIS and questionnaires at home. Though instruction to the test clearly stated that the participant should complete the survey as independently as possible, it was not possible to rule out that some (especially children) received too much help from their parents. 

## 5. Conclusions

BY-GIS is a novel experimental design to assess biases of interpretation and memory toward gastrointestinal stimuli in children and adolescents. This was a first test and exploration of potential gender- and age-related differences in a healthy normative sample to derive comparison data on bias toward gastrointestinal-related stimuli. We found few significant differences between gender and age groups. Overall, the results support BY-GIS as suitable within the age span. In future research, BY-GIS could be used to assess cognitive biases in children and adolescents with FAPD with data from the present study as comparison material. Furthermore, BY-GIS could be used as a framework to assess cognitive biases in children and adolescents with other clinical disorders by exchanging the present stimuli (words and pictures) with disorder-specific stimuli for other conditions.

## Figures and Tables

**Figure 1 children-10-01327-f001:**
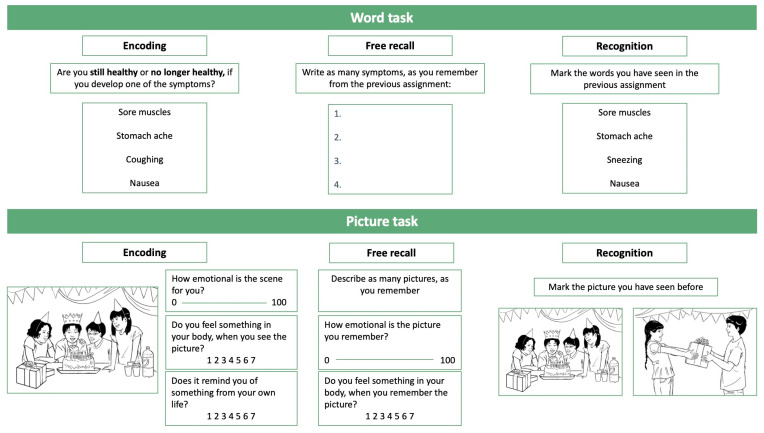
Overview of the experimental word task and picture task design in BY-GIS. Pictures are reprinted from the Pictures with Social Context and Emotional Scenes (PiSCES) database with permission from the publisher of [31]. 2023, Springer Nature.

**Figure 2 children-10-01327-f002:**
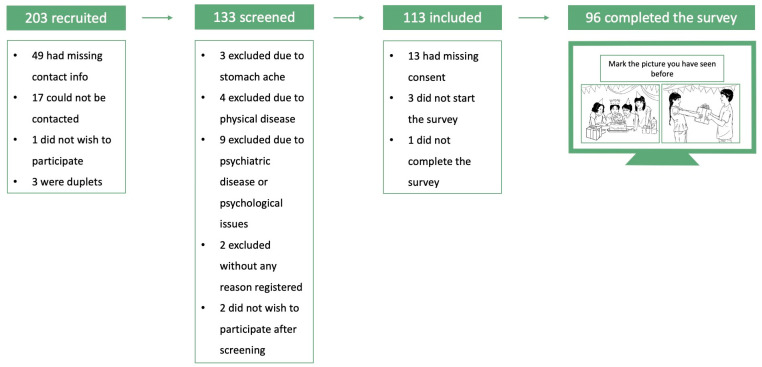
Flowchart of the recruiting and inclusion process. Pictures are reprinted from the Pictures with Social Context and Emotional Scenes (PiSCES) database with permission from the publisher of [31]. 2023, Springer Nature.

**Table 1 children-10-01327-t001:** Sample characteristics of participants and their parents.

Participants	Possible Range	Total	Children (8 to 12 Years)	Adolescents (13 to 17 Years)
N (% female)		96 (47.92%)	51 (49.02%)	45 (46.67%)
Age (mean (SD))		12.32 (2.56)	10.27 (1.43)	14.64 (1.23)
Scores on questionnaires (median (IQR))				
Children’s Somatic Symptoms Inventory ^1^	0–96	8.00 (11.00)	8.00 (11.0)	11.0 (10.00)
Childhood Illness Attitude Scale *^2^*	11–33	15.00 (5.00)	14.00 (6.00)	16.00 (4.00)
Pediatric Quality of Life Inventory ^1^, Gastrointestinal Symptoms scale	0–100	86.11 (19.44)	91.66 (19.44)	86.11 (16.66)
Parents				
Education				
High school, N (%) ^3^		<5	<5	<5
Vocational education, N (%)		13 (13.54)	6 (11.76)	7 (15.56)
Continuing education, N (%) ^3^		8 (8.33)	<5	5 (11.11)
Higher education, N (%)		74 (77.08)	42 (82.35)	32 (71.11)
Employment				
Full- or part-time employment, N (%)		89 (92.71)	45 (88.24)	44 (97.78)
Unemployed, N (%) ^3^		<5	<5	<5
Other (e.g., sick leave, maternity leave), N (%) ^3^		6 (6.25)	5 (9.80)	<5
Marital status				
Living together, N (%)		85 (85.54)	46 (90.20)	39 (86.67)
Living apart, N (%)^3^		10 (10.42)	<5	6 (13.33)
Other, N (%) ^3^		<5	<5	<5
Yearly household income				
Low income (<500,000 DKK), N (%)^3^		5 (5.21)	<5	<5
Middle income (500,000–1,000,000 DKK), N (%)		47 (48.96)	24 (47.06)	23 (51.11)
High income (>1,000,000 DKK), N (%)		44 (45.84)	23 (45.10)	21 (46.67)

^1^ Only 94 participants completed the Children Somatic Symptoms Inventory and Pediatric Quality of Life Inventory questionnaires. ^2^ Only 95 participants completed the Childhood Illness Attitude Scale questionnaire. ^3^ Numbers between 0 and 5 are denoted as “<5” due to data protection rules in Denmark. There was no significant difference between children and adolescents for the Children’s Somatic Symptoms Inventory, Childhood Illness Attitude Scale or the Pediatric Quality of Life Inventory Gastrointestinal symptoms scale and no significant differences in education, employment, marital status, or yearly household income between the parents of children and adolescents (all *p*-values > 0.05) (Wilcoxon rank-sum test). DKK = Danske kroner (Danish currency). IQR = Interquartile Range. SD = Standard deviation.

**Table 2 children-10-01327-t002:** Median (IQR) scores within the three phases for the word task.

	Possible Range	Total (N = 96)	Girls (N = 46)	Boys (N = 50)	Test statistic	*p*	Children (N = 51)	Adolescents (N = 45)	Test Statistic	*p*
Encoding										
Healthy		12.50 (5.00)	13.00 (4.00)	12.00 (6.00)	0.45 ^b^	0.65	12.00 (6.00)	13.00 (4.00)	−1.59 ^b^	0.11
No longer healthy		7.50 (5.00)	7.00 (4.00)	8.00 (6.00)	−0.45 ^b^	0.65	8.00 (6.00)	7.00 (4.00)	1.59 ^b^	0.11
Free recall										
Words recalled	0–20	7.00 (6.00)	8.00 (6.00)	6.50 (5.00)	0.85 ^b^	0.40	6.00 (4.00)	8.00 (4.00)	−2.17 ^b^	0.03
Gastrointestinal words recalled	0–10	3.00 (3.00)	3.00 (3.00)	3.00 (3.00)	0.28 ^b^	0.78	3.00 (3.00)	3.00 (3.00)	−0.42 ^b^	0.67
General words recalled	0–10	4.00 (3.00)	4.00 (4.00)	3.00 (4.00)	1.16 ^b^	0.25	3.00 (3.00)	4.00 (5.00)	−3.31 ^b^	<0.01
Recognition										
Words correctly recognized in total	0–40	37.00 (2.00)	37.00 (2.00)	37.00 (2.00)	1.01 ^a^	0.31	37.00 (2.00)	37.00 (3.00)	−0.75 ^a^	0.45
General words correctly recognized	0–20	19.00 (2.00)	19.00 (1.00)	19.00 (2.00)	1.60 ^a^	0.11	19.00 (1.00)	19.00 (2.00)	1.28 ^a^	0.20
Difficulty	1–7	2.00 (2.50)	2.50 (2.00)	2.00 (3.00)	0.49 ^a^	0.63	2.00 (2.00)	2.00 (3.00)	0.59 ^a^	0.55
Confidence	1–7	5.00 (2.00)	5.00 (2.00)	5.00 (2.00)	−0.23 ^a^	0.82	5.00 (1.00)	5.00 (2.00)	1.91 ^a^	0.06
Gastrointestinal words correctly recognized	0–20	18.00 (2.00)	18.00 (2.00)	18.50 (2.00)	−0.16 ^a^	0.87	18.00 (3.00)	19.00 (2.00)	−2.17 ^a^	0.03
Difficulty	1–7	3.00 (2.00)	3.00 (1.00)	3.00 (2.00)	−0.39 ^a^	0.70	3.00 (2.00)	3.00 (1.00)	1.14 ^a^	0.26
Confidence	1–7	5.00 (2.00)	5.00 (2.00)	5.00 (2.00)	−0.19 ^a^	0.85	5.00 (2.00)	5.00 (2.00)	0.53 ^a^	0.60

^a^ Wilcoxon rank-sum test used due to non-normal distribution. For Wilcoxon rank-sum test “z-value” is reported. ^b^ Students *t*-test used due to normal distribution. The “*t*-value” is reported.

**Table 3 children-10-01327-t003:** Median (IQR) scores within the three phases for the picture task.

	Possible Range	Total (N= 96)	Girls (N = 46)	Boys (N = 50)	Test Statistic	*p*	Children (N = 51)	Adolescents (N = 45)	Test Statistic	*p*
Encoding										
Emotional valence	1–100	69.00 (11.40)	69.60 (10.06)	68.50 (10.33)	−1.30 ^b^	0.20	67.93 (13.00)	70.00 (8.80)	−1.02 ^b^	0.31
Physical reaction	1–7	1.27 (0.93)	1.40 (1.07)	1.20 (0.93)	1.22 ^a^	0.22	1.47 (1.33)	1.20 (0.46)	2.05 ^a^	0.04
Self-relevance	1–7	4.37 (1.30)	4.13 (1.27)	4.43 (1.27)	−0.49 ^b^	0.63	4.20 (1.27)	4.40 (1.33)	−1.01 ^b^	0.31
Free recall										
Pictures recalled	0–15	3.00 (2.00)	3.00 (2.00)	3.00 (3.00)	0.30 ^b^	0.76	3.00 (2.00)	3.00 (2.00)	−0.74 ^b^	0.46
Fun pictures recalled	0–5	1.00 (2.00)	1.00 (2.00)	1.00 (2.00)	−0.43 ^b^	0.67	1.00 (2.00)	1.00 (1.00)	0.18 ^b^	0.86
School pictures recalled	0–5	1.00 (2.00)	1.00 (2.00)	1.00 (2.00)	0.39 ^b^	0.70	1.00 (2.00)	1.00 (2.00)	−1.11 ^b^	0.27
Food pictures recalled	0–5	1.00 (1.50)	1.00 (1.00)	1.00 (2.00)	0.54 ^b^	0.59	1.00 (2.00)	1.00 (1.00)	−0.33 ^b^	0.75
Emotional valence of recalled picture	1–100	68.20 (20.76)	64.50 (16.25)	74.38 (19.75)	−2.43 ^b^	0.01	68.40 (24.64)	68.00 (20)	−0.41 ^b^	0.68
Physical reaction of recalled pictures	1–7	1.20 (1.00)	1.45 (1.25)	1.00 (1)	1.43 ^a^	0.15	1.60 (1.71)	1.10 (0.50)	2.05 ^a^	0.04
Recognition										
Pictures correctly recognized	0–15	15.00 (0)	15.00 (0)	15.00 (0)	1.95 ^a^	0.05	15.00 (0)	15.00 (0)	0.11 ^a^	0.92
Difficulty	1–7	1.07 (0.27)	1.07 (0.13)	1.20 (0.33)	−1.34 ^a^	0.18	1.07 (0.33)	1.07 (0.20)	1.32 ^a^	0.19
Confidence	1–7	6.93 (3.00)	6.97 (0.47)	6.93 (0.20)	0.02 ^a^	0.98	6.93 (0.73)	6.93 (0.13)	−1.32 ^a^	0.19

^a^ Wilcoxon rank-sum test used due to non-normal distribution. For Wilcoxon rank-sum test, “z-value” is reported. ^b^ Student’s *t*-test used due to normal distribution. For Student’s *t*-test, “t-value” is reported.

## Data Availability

The data used in the present study contain sensitive personal information and therefore cannot be shared freely due to Danish data protection laws. All data are stored in REDCap, and access can only be granted with approval from the Central Region, Denmark, which has legal responsibility for data as the data manager. If access is granted, the principal investigator and last author (Charlotte Ulrikka Rask, charrask@rm.dk) will make data available. The General Data Protection Regulation (GDPR) and the Danish Data Protection Act prohibit any other forms of data sharing.

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
