# Peer review of "A New Experimental Design to Examine Cognitive Biases for Gastrointestinal Related Stimuli in Children and Adolescents"

_children, 2023, doi:10.3390/children10081327_

Round 1

Reviewer 1 Report

The authors aimed to develop a new experimental design to assess cognitive biases toward gastrointestinal-related stimuli in children and adolescents. This tool is to be used in patients with FAPD. This is one of the only studies performed in children and adolescents, as studies analyzing the cognitive biases in functional disorders are made mainly in adults.

The paper is well written, with very few issues to be improved. I would suggest adding more data to the abstract. Besides this aspect, the introduction, methods, results, and discussions are appropriate. The conclusions were supported by the data presented. Also, the authors included a paragraph with the strengths and limitations of their study. Overall, this is a good study analyzing an interesting issue of functional disorders in children and adolescents. 

There are no important issues regarding the English language, but only some editing improvements (abbreviated words that should be explained - HNST, PiSCES).

Author Response

The authors aimed to develop a new experimental design to assess cognitive biases toward gastrointestinal-related stimuli in children and adolescents. This tool is to be used in patients with FAPD. This is one of the only studies performed in children and adolescents, as studies analyzing the cognitive biases in functional disorders are made mainly in adults.

The paper is well written, with very few issues to be improved. I would suggest adding more data to the abstract. Besides this aspect, the introduction, methods, results, and discussions are appropriate. The conclusions were supported by the data presented. Also, the authors included a paragraph with the strengths and limitations of their study. Overall, this is a good study analyzing an interesting issue of functional disorders in children and adolescents. 

Response:

Thank you very much for your positive feedback, this is highly appreciated. We agree that more data should be added to the abstract. Accordingly, we have performed the following additions in the abstract, line 29-32.

There are no important issues regarding the English language, but only some editing improvements (abbreviated words that should be explained - HNST, PiSCES).

Response:

Thank you pointing out the missing explanations of our abbreviations, these are clearly explained now.

Reviewer 2 Report

I commend the authors on putting together this tool.

However, until it has been assessed among youth with FAP with the current group data as controls, the current data and the tool itself have very limited relevance. The target population need to assess feasibility, satisfaction etc and then between group differences assessed to determine efficacy.

The entire manuscript is in landscape format, not portrait.

You methods section needs to be broken up with sub-headings and bullet points to make easier to read and understand. The discussion is frequently a repeat of results with little discussion of how your results fit in with the wider literature.

The results need to be greatly simplified – use sub-headings and consider using figures

Table 2 is very busy – remove the decimal points from those with ‘.00’ and it will be much easier to read.

Many of your references are incomplete so not possible to retrieve the literature you have cited.

The literature cited does not seem particularly up to date but the year is missing from many so impossible to assess.

English language was acceptable but there is an over-use of commas.

Author Response

I commend the authors on putting together this tool. However, until it has been assessed among youth with FAP with the current group data as controls, the current data and the tool itself have very limited relevance. The target population need to assess feasibility, satisfaction etc and then between group differences assessed to determine efficacy.

Response:

Thank you for your very helpful and relevant comments. We have addressed each of them in our response below. We believe the present study is an important step towards increasing the knowledge on cognitive biases in children and adolescents. As stated the focus of the paper is to describe in more detail the development of a specific experimental paradigm for this age group as such studies are scarce in the literature. Further, we have tested the feasibility of the paradigm on healthy young people within the same age group as the proposed target group. Thereby, we hope to ensure that data collection on the target group will be feasible across gender and age span and not wasted due to the application of an experimental design unsuitable for young people. Nevertheless, we do agree that the inclusion of a clinical sample could have added further value to the study, and this has now been addressed in the "Strengths and limitations" section, line 441-445.

The entire manuscript is in landscape format, not portrait.

Response:

We agree that the manuscript should be in portrait form, but it has been changed to vertical during the submission process. It should now be in portrait form except figures and Table 2 and 3.

You methods section needs to be broken up with sub-headings and bullet points to make easier to read and understand.

Response:

The Methods section has been further divided with subheadings and numbers. See line 110-115, line 144-145, line 175-180 and line 241-246.

The discussion is frequently a repeat of results with little discussion of how your results fit in with the wider literature.

Response:

Thank you for your relevant comment on the discussion. We have tried elaborating the discussion by comparing to other studies investigating youth's performance in experimental paradigms, see line 339-344. Further, we have made changes of discussion on line 339-348, 358-361, 362-372, 378-392, 403-405, 417-424. If you have any further specific suggestions, we will gladly consider them for further improvement of the discussion.

The results need to be greatly simplified – use sub-headings and consider using figures.

Response:

The results section has been simplified by dividing it into subheadings, see line 268-273, 274-280, 281-285, 307-312, 313-323 and 324-327.

Table 2 is very busy – remove the decimal points from those with ‘.00’ and it will be much easier to read.

Response: We agree that Table 2 (and Table 3) can seem busy. However, we believe that the same decimal points should be used throughout the whole paper. Therefore, we have kept the two decimals as we think one decimal will not be sufficient. Instead, we have made the tables wider, reduced and aligned the text to the left and marked significant p-values. We hope that these changes will improve the readability of the tables. If this is not the case, we will gladly consider changing them further.

Many of your references are incomplete so not possible to retrieve the literature you have cited. The literature cited does not seem particularly up to date but the year is missing from many so impossible to assess.

Response:

We apologize for the incomplete references. We have now updated these incomplete references (7, 9, 10, 20, 21, 33, 36 and 46) and reference 3, 17 and 28 have replaced older references, so the reference list is more up to date. Further, we have added reference 42, 43, 44 and 45.

English language was acceptable but there is an over-use of commas.

Response: 

The text has been revised by a professor secretary with a Master's degree in English. We hope this have eliminated the incorrect commas.

Reviewer 3 Report

Dear Author,

Thank you for the opportunity to read and evaluate the article. It is a well-written article in general, but there are some shortcomings to take it higher.

1- The abbreviation in parentheses should be removed from the title.

2- Abstract: a- It is recommended to arrange it in the form of abstract, introduction, method, findings and conclusion.

                       b- The start and end dates of the study should be added.

                       c- Findings should be strengthened by adding more numbers and p values to the part.

3- Too many abbreviations throughout the article make it difficult to read, it should be reduced.

4- The discussion should be expanded.

5- The sources before 2010 should be updated.

Minor editing of English language required

Author Response

Thank you for the opportunity to read and evaluate the article. It is a well-written article in general, but there are some shortcomings to take it higher.

Response:

Thank you very much for your positive review of our article. Your comments have been very helpful in the further improvement of our paper.

1- The abbreviation in parentheses should be removed from the title.

Response:

The abbreviation is now removed and introduced in the abstract instead.

2- Abstract: a- It is recommended to arrange it in the form of abstract, introduction, method, findings and conclusion.

                       b- The start and end dates of the study should be added.

                       c- Findings should be strengthened by adding more numbers and p values to the part.

Response:

  1. a) According to the journal's author guidelines the abstract is not to have headings, which is the reason for us not to include these.
  2. b) This is now added to the abstract, see line 28.
  3. c) We have expanded the section on results and included P values. See line 29-31. However, due to a maximum word count of 200 we find it difficult to add further numbers and information.

3- Too many abbreviations throughout the article make it difficult to read, it should be reduced.

Response:

Thank you for this helpful insight. The abbreviations have now been reduced and hopefully this improves the overall readability of the paper.

4- The discussion should be expanded.

Response:

This point was also made by one of the other reviewers. We hope that the additions we have made, including a further discussion of the study in relation to the existing literature, has improved it sufficiently. Elaboration of the discussion can be seen on the following lines 339-348, 358-361, 362-372, 378-392, 403-405, 417-424.

5- The sources before 2010 should be updated.

Response:

We agree that some of the referenced literature is older and have included more recent sources when possible. Specifically, reference 3, number 17 and number 28 has replaced previous references. However, some of the references from before 2010 concerns original paradigms that inspired our work. We have therefore kept these in the reference list as well.

Round 2

Reviewer 2 Report

Thank you for making the revisions in response to my first round of comments. However, my original point stands that without testing feasibility among the target population your paper is methodologically flawed and publishing as is, without testing among the target cohort, is misleading. While you have acknowledged this is a limitation it is not sufficient for me to endorse publication. Expanding the paper with a cohort of children with FAP would make this an extremely important contribution to the treatment of children with FGID, but this cohort is essential for inclusion.

Author Response

Author's reply:

Thank you very much for your feedback and positive words on our study. As suggested by the editor we have now made some additional changes that further speaks to your relevant point that it could have been a further strength to have included the target group for a feasibility testing. Importantly, we have now changed the wording of aim 2 as stated above to: "To derive comparative data on bias to gastrointestinal stimuli using a healthy "normative" sample."

Reviewer 3 Report

Dear Authors,

It is seen that the authors made the changes requested by the referees. After this stage, I think there is no obstacle in accepting the article. Thanks to the authors.

Kind regards.

Author Response

Author's reply:

Thank you very much for this positive feedback, this is highly appreciated!